

# Evolution of African barbs from the Lake Victoria drainage system, Kenya

Violet M. Ndeda[1,2], Mariana Mateos[1] and Luis A. Hurtado[1]

[1] Wildlife and Fisheries Sciences, Texas A&M University, College Station, TX, United States of America
[2] Department of Zoology, Maseno University, Maseno, Kenya

## ABSTRACT

The Lake Victoria drainage basin (LVD) in Kenya is home to ten nominal species of small barbs (*Enteromius*) and one of large barbs (*Labeobarbus altianalis*). A recent molecular study genetically characterized small barbs in this region and found evidence of introgression between certain species, complicating the taxonomy and species identification of these fishes. This study aimed to extend our understanding on the evolution of these fishes by: (1) determining whether putatively pure individuals of *Enteromius cercops* are found in the Kenyan LVD, as the previous study only found hybrid individuals of this species in this region; (2) testing the sister relationship between *Enteromius profundus*, endemic to Lake Victoria, and *Enteromius radiatus*, also found in Lake Victoria, which had been previously synonymized; (3) examining the phylogenetic relationships of small barbs of the Kenyan LVD with those reported from other ichthyological provinces of Africa; and (4) examining the phylogenetic relationships of *Labeobarbus altianalis* with other *Labeobarbus* species. To this end, we obtained mitochondrial Cytochrome b and nuclear Growth Hormone (GH) intron 2 gene sequences of nine *Enteromius* species from the LVD in Kenya, as well as cytochrome b sequences for *L. altianalis*. We conducted Maximum likelihood and Bayesian phylogenetic analyses to establish their evolutionary relationships in relation to many other barbs specimens from Africa. Phylogenetic analyses did not reveal instances of hybridization/introgression among the individuals sequenced by us. A sister relationship between *E. profundus* and *E. radiatus* was not found. This latter species shows instead a sister relationship with a lineage comprised of two species from West Africa. Other sister relationships between taxa from the East coast and other ecoregions from Africa are observed, suggesting that past drainage connections and vicariant events contributed to the diversification of *Enteromius*. Finally, only a single haplotype was recovered among the *L. altianalis* individuals examined, which is most similar to a specimen from Lake Edward in Uganda.

# INTRODUCTION

Barbs constitute a significant component of the freshwater fish fauna of Africa, and represent the most species-rich group of cyprinids on this continent (*Hayes & Armbruster, 2017*; *Leveque & Daget, 1984*; *Ren & Mayden, 2016*; *Skelton, 1988*; *Skelton, 1993*; *Skelton, Tweddle & Jackson, 1991*). Molecular characterization of African barbs from different

Corresponding authors
Mariana Mateos,
marianamateosh@gmail.com,
mmateos@tamu.edu
Luis A. Hurtado, lhurtado@tamu.edu

regions has greatly contributed to our understanding on the diversity and evolution of these fishes (*Beshera, Harris & Mayden, 2016*; *De Graaf et al., 2007*; *Hayes & Armbruster, 2017*; *Muwanika et al., 2012*; *Ren & Mayden, 2016*; *Schmidt, Bart & Nyingi, 2017*; *Yang et al., 2015*). A large dataset of DNA sequences (particularly of the mitochondrial Cytochrome b gene) of African barbs from different regions has accrued, providing a resource for performing phylogenetic analyses across regions, which will enhance knowledge on the systematics, evolution, and biogeographic history of this important group.

Although they were treated as part of *Barbus* Cuvier and Cloquet, 1816, which included >800 species distributed across Eurasia and Africa (*Berrebi et al., 1996*; *Skelton, 2001*; *Skelton, Tweddle & Jackson, 1991*), molecular phylogenetic studies have corroborated that this taxonomically complex and heterogeneous assemblage is a polyphyletic group (*Ren & Mayden, 2016*; *Tsigenopoulos et al., 2002*; *Yang et al., 2015*). The large hexaploid African barbs are now classified as *Labeobarbus* (tribe Torini), with ca. 125 recognized species (*Tsigenopoulos, Kasapidis & Berrebi, 2010*; *Vreven et al., 2016*; *Yang et al., 2015*). The large tetraploid African barbs are classified within the tribe Smiliogastrini (*Yang et al., 2015*), comprising five genera: *Pseudobarbus*; *Cheilobarbus*; *Amatolacypris*; *Sedercypris*; *Namaquacypris* (*Skelton, Swartz & Vreven, 2018*). The small, diploid, African barbs have also been assigned to the tribe Smiliogastrini, and *Yang et al. (2015)* proposed to include all of them within the genus *Enteromius*, the oldest available genus-group name for these fishes, even though they do not appear to correspond to a monophyletic group. This proposal is controversial, with some authors supporting it (*Hayes & Armbruster, 2017*), whereas others proposing that this group be referred to as '*Barbus*' to reflect its taxonomic uncertainty (*Schmidt & Bart Jr, 2015*; *Stiassny & Sakharova, 2016*). In this study, we refer to them as *Enteromius*, even though a recent study (*Schmidt, Bart & Nyingi, 2017*) that was conducted in the same area as the present study used '*Barbus*'. According to *Hayes & Armbruster (2017)*, *Enteromius* includes 218 nominal species.

Barbs are an important biodiversity component of the Lake Victoria drainage Basin (LVD) in Kenya, and play a significant role in food security and socioeconomic development of the local community (*Ochumba & Many Ala, 1992*; *Okeyo, 2014*). A recent multilocus study (*Schmidt, Bart & Nyingi, 2017*), molecularly characterized most nominal species of small barbs present in this region. Phylogenetic relationships among them were analyzed, and high levels of genetic divergence within some recognized species were uncovered. Further complicating the taxonomy and species identification within this group, this study revealed evidence of introgression/hybridization involving five small barb species. Several important phylogenetic questions, however, still remain to be answered for Kenyan LVD barbs.

First, due to mitochondrial introgression from other species (i.e., *E. neumayeri* or *E.* c.f. *paludinosus* Jipe), *Schmidt, Bart & Nyingi (2017)* were not able to obtain Cytb sequences that could be attributed to the *E. cercops* lineage. It is thus unclear whether pure populations of this species exist in the Kenyan LVD, which is important for conservation. Lack of Cytb sequences also limits examination of the evolutionary relationships of *E. cercops* with the other small African barbs for which Cytb sequences are available.

Second, *Schmidt, Bart & Nyingi (2017)* did not include *Enteromius profundus* in their study, a species endemic to Lake Victoria, for which molecular analyses can help to clarify its evolutionary history and taxonomy. *Greenwood (1970)* originally described *E. profundus* as a subspecies of *E. radiatus*, another species found in Lake Victoria and other localities in Kenya. He considered *E. radiatus* to be comprised of three subspecies: *B. radiatus profundus*, *B. radiatus radiatus* and *B. radiatus aurantiacus*. *Stewart (1977)*, however, based on meristic and morphometric analyses, concluded that *E. profundus* is a separate species from *E. radiatus*; but did not find a basis for the separation of the other two subspecies. The two species occupy different depths in Lake Victoria; *E. profundus* is distributed at depths between 16 and 65 m (*Greenwood, 1970*), whereas *E. radiatus* occupies shallower waters (*Stewart, 1977*). Molecular analyses are thus needed to examine the relationship between *E. profundus* and *E. radiatus*.

Third, phylogenetic relationships of small barbs from this region, which belong to the East Coast province, with those from other African ichthyological provinces have not been examined. Numerous sequences, mostly of the Cytb gene, are available for other small barbs from the East Coast, as well as the Nilo-Sudan, Upper Guinea, Lower Guinea, Congo, and Southern provinces (as defined by *Levêque et al., 2008*; *Roberts, 1975*). The African continent has had a complex and dynamic geological history, in which past hydrological connections may have enabled exchange of taxa from different regions (*Salzburger, Van Bocxlaer & Cohen, 2014*; *Stewart, 2001*).

Finally, Cytb sequences for the large barb *Labeobarbus altianalis* in the Kenyan LVD have not been examined. This species has historically constituted an important fishery in this region (*Whitehead, 1959*), but overfishing has severely decimated its populations (*Ochumba & Many Ala, 1992*). Genetic diversity for this species in the Kenyan LVD has been studied with the mitochondrial control region, which revealed some population structure (*Chemoiwa et al., 2013*). *Muwanika et al. (2012)* examined partial Cytb sequences for *L. altianalis* from different localities in Uganda, including the Lake Victoria and Albertine basins. Therefore, obtaining Cytb sequences from this species in the Kenyan LVD will allow examination of differences among the populations from both countries. In addition, a large dataset of Cytb sequences exists for *Labeobarbus* from different regions in Africa, which has not been analyzed with this species (*Beshera, Harris & Mayden, 2016*).

Herein we obtained Cytb sequences from eight species of small barbs and the large barb *L. altianalis* from different localities in the Kenyan LVD, and conducted phylogenetic analyses of these with a large dataset of reported sequences of small and large African barbs. We also obtained sequences of the nuclear GH intron and conducted phylogenetic analyses to help determine whether individuals were pure or exhibited evidence of hybridization/introgression (*Schmidt, Bart & Nyingi, 2017*). Our main objectives were to: (1) determine whether putatively pure individuals of *E. cercops* are found in the Kenyan LVD; (2) test the sister relationship between *E. profundus* and *E. radiatus*; (3) examine the phylogenetic relationships of small barbs of the Kenyan LVD with those reported from other ichthyological provinces of Africa; and (4) examine the phylogenetic relationships of *L. altianalis* with other *Labeobarbus* species.

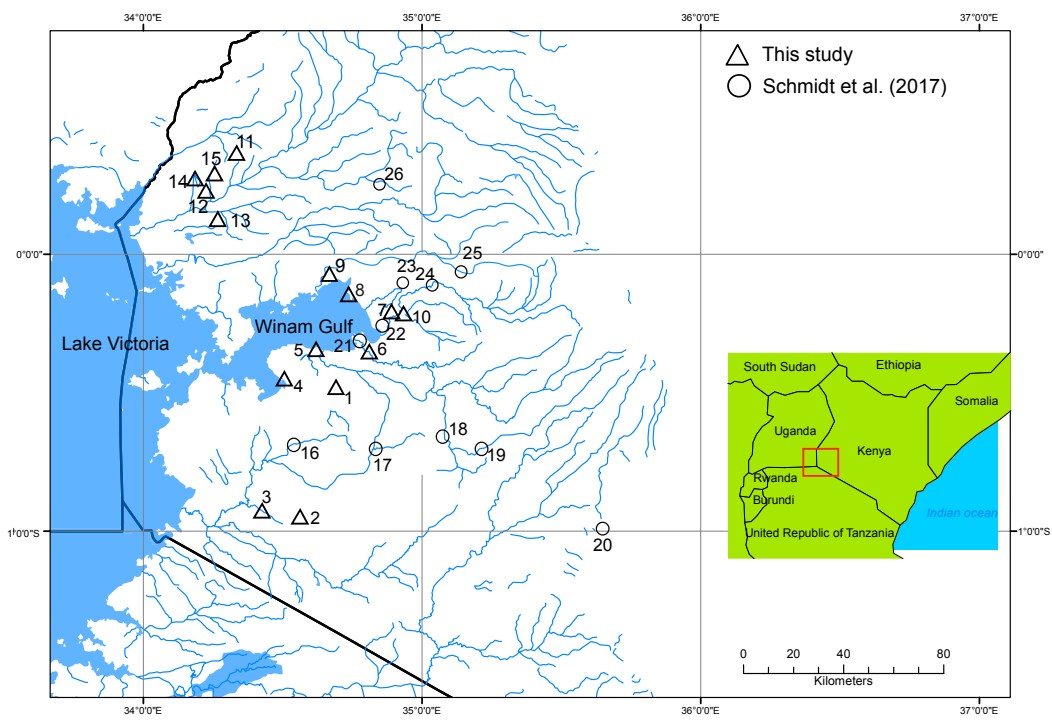

**Figure 1  Localities of specimens sampled in this study (triangles).** Circles represent localities within the Lake Victoria Drainage (LVD) sampled by *Schmidt, Bart & Nyingi (2017)* that were not sampled in our study (locations are approximate based on their description of locality, as coordinates were not reported). The map was developed with ArcMap version 10.3—a part of the ESRI ArcGIS® Desktop suite. Localities where each species was sampled for our study are as follows: *Enteromius apleurogramma* (2); *Enteromius cercops* (1, 5, 7, 9); *Enteromius* cf. *paludinosus* (13); *E. jacksoni* (7); *E. kerstenii* (1, 6, 14); *E. neumayeri* (3); *E. nyanzae* (7, 10); *E. profundus* (8); and *L. altianalis* (4, 5, 10, 11, 12).

## MATERIALS AND METHODS

### Tissue source for DNA

We used ethanol-preserved fin clips from nine species (*L. altianalis, E. apleurogramma, E. profundus, E. cercops, E. nyanzae, E. kerstenii, E. jacksoni, E. neumayeri,* and *E. paludinosus*) loaned by the Kenya Marine and Fisheries Institute (KMFRI). The remainder of the specimens is stored in formalin at KMFRI. These specimens were originally identified by fish taxonomists from KMFRI using morphological identification keys according to *Greenwood (1962)*. They were obtained from sixteen localities in the Lake Victoria drainage area (LVD) in Kenya, which included Lake Victoria, rivers draining to the lake and associated dams (represented by triangles in Fig. 1; circles indicate the approximate location of specimens from *Schmidt, Bart & Nyingi (2017)* included in our analyses).

### DNA isolation, PCR amplification and sequencing

Genomic DNA was extracted using the DNeasy Blood and Tissue Kit (QIAGEN Inc., Valencia, CA, USA). The quality of extracted DNA was examined by visualization

on a 1.5% agarose electrophoresis gel, and quantified with a NanoDrop® ND-1000 spectrophotometer. Fragments of one mitochondrial (Cytochrome *b*; Cytb; ~1,140 bp) and one nuclear (Growth Hormone Intron 2; GH; ~520 bp) gene were PCR amplified from 1–4 individuals per locality. PCR was performed in a 25 μl reaction containing 19.9 μl ultrapure water, 0.5 μl dNTP mix (2.5 mM), 2.5 μl of 10X buffer, 0.5 μl of each 10 μM primer, 0.1 μl *Taq* polymerase (OneTaq; New England Biolabs, Inc., Ipswich, MA, USA), and 1 μl of DNA template. Cytb was amplified with primers Cytb L15267 (5′AATGACTTGAAGAACCACCGT3′) and H16461 (5′CTTCGGATTACAAGACC3′), following *Briolay et al. (1998)*. GH intron 2 was amplified using primers GH102F (5′TCGTGTACAACACCTGCACCAGC-3′), GH148R (5′ TCCTTTCCGGTGGGTGCCTCA-3′), from *Mayden et al. (2009)*. PCR amplification included a denaturation step of 2 min at 95 °C followed by 35 cycles of 1 min at 95 °C, 30 s at 58–60 °C (Cytb)/ 55 °C (GH) and 1 min at 72 °C followed in turn by a final extension of 6 min at 72 °C. Successful amplification was verified by running the PCR amplicons alongside a standard Lambda ladder on a 1.5% agarose gel stained with GelRed™ (Biotium Inc., Hayward, CA, USA). Products were sequenced bi-directionally using the amplification primers in an ABI 3730 capillary sequencer.

## Sequence assembly and alignment

Nucleotide sequences were assembled and edited with Sequencher 4.8 (Gene Codes, Ann Arbor, MI, USA). Newly generated Cytb of *Enteromius* were combined with publicly available sequences of African *Enteromius* and their allies (e.g., *Systomus*, *Barboides*, *Clypeobarbus*, *Pseudobarbus*, *Labeobarbus*), including sequences from *Schmidt, Bart & Nyingi (2017)*. Sequences were aligned with MAAFT v.6.0 (*Katoh & Toh, 2008*). Cytb aligned sequences were translated into amino acids to verify the alignments and to rule out the occurrence of frameshifts and early stop codons that could be indicative of pseudogenes or sequencing errors. Species from the family Catostomidae, which represent a group of tetraploids thought to have arisen due to a hybridization event early (60 million years) in the history of the cypriniform fishes, were initially used as outgroups (*Uyeno & Smith, 1974*). Following preliminary analyses, the dataset was pruned to retain only taxa relevant to this study: the newly generated sequences; small barbs closely related to the taxa in this study; close relatives of *L. altianalis*, and four appropriate outgroup taxa (*Pethia ticto*, *Hampala macrolepidota*, *Puntigrus tetrazona*, and *Systomus sarana*; following *Schmidt, Bart & Nyingi (2017)*. The GH dataset included the newly generated sequences, representatives of seven *Enteromius* species from the same region, and sequences of *Pethia* and *Garra* were used as outgroups.

## Phylogenetic analyses

Phylogenetic analyses were performed using maximum likelihood (*Stamatakis, 2014*), and Bayesian inference (*Huelsenbeck et al., 2001*). Appropriate models of sequence evolution for these analyses were determined using PARTITIONFINDER v2.7 (*Guindon et al., 2010*; *Lanfear et al., 2012*; *Lanfear et al., 2017*) and JModeltest 2.1.10 (*Darriba et al., 2012*) under the Akaike Information Criterion (AIC), corrected AIC(c), and Bayesian Information Criterion (BIC). Best models found and partition schemes tested are provided in Table 1.

**Table 1  Description of characters and substitution models identified by model selection analyses.**  Best model selected by: (a) jModeltest 2.1.10 v20160303 (*Darriba et al., 2012*) according to each criterion (AICc, AIC, BIC) and its corresponding weight (on a fixed BioNJ tree); and (b) the best partitioning scheme according to the BIC implemented in PartitionFinder 2.7 (*Guindon et al., 2010*; *Lanfear et al., 2012*; *Lanfear et al., 2017*): branchlengths = linked; and search = greedy.

| Gene | Non-redundant taxa | Characters used | Parsimony informative | Partitioning Scheme | AICc (weight) | AIC (weight) | BIC (weight) |
|------|------|------|------|------|------|------|------|
| **GH** | 46 | 201 | 63 | 1 | TPM2uf (0.36) | TVM (0.22) | TVM (0.24) |
| **Cytb** | 291 | 1,023 | 507 | 1 | TIM2+I+G (0.99) | TIM2+I+G (0.61) | TIM2+I+G (0.99) |
| | | | | 3 (by codon) | | | |
| | | | | Codon 1 | | | SYM+I+G |
| | | | | Codon 2 | | | HKY+I+G |
| | | | | Codon 3 | | | GTR+G |

Bayesian analyses were performed in MrBayes 3.2.6 (*Ronquist et al., 2012*) via the CIPRES Science Gateway (*Miller, Pfeiffer & Schwartz, 2010*). We used the model indicated by the BIC criterion of JModeltest or the closest more complex model available in MrBayes. The analysis was run for 10,000,000 generations consisting of four independent Markov Chain Monte Carlo (MCMC) chains sampled at every 1,000 generations. TRACER v1.6 was used to assess MCMC stationarity and to ensure adequate effective sampling size values (>200) were achieved. The first 25% of the sampled trees were discarded as burn-in, whereas the remaining sampled trees were summarized with "sumt" command implemented in MrBayes.

Maximum Likelihood (ML) analysis was implemented in RAxML v 8.2.6 (*Stamatakis, 2014*) using rapid bootstrap and GTRGAMMA model via the CIPRES Science Gateway (*Miller, Pfeiffer & Schwartz, 2010*) to generate a maximum likelihood tree. Clade support was examined by a nonparametric bootstrap analysis of 200 replicates and summarized with 50% majority rule consensus tree computed using the SUMTREES script (v.3.3.1) (*Sukumaran & Holder, 2010*).

Kimura-2-parameter (K2P) pairwise genetic distances were obtained in Paup v. 4a(build 161) (*Swofford, 2002*), ignoring sites with missing data or gaps (i.e., option "missDist=ignore").

## RESULTS AND DISCUSSION

We obtained new Cytb sequences for 48 specimens and new GH sequences for 34 specimens (overlap = 26; see Table S1; GenBank Accession Nos. MH484522–MH484603). None of the specimens had GH chromatogram peak patterns suggestive of heterozygosis. Alignments are available as nexus files under Data S1 and S2. Phylogenetic reconstructions using GH and Cytb DNA sequences are shown in Figs. 2, 3–8, respectively. Regarding species names, although we acknowledge that there are many misidentifications of the taxa available on GenBank (*Hayes & Armbruster, 2017*), we retain the species name used by the original contributor of the corresponding sequence. With the exception of *E. cercops* Cytb sequences (see below), for the seven *Enteromius* species that overlapped between our study and that of *Schmidt, Bart & Nyingi (2017)* within the LVD region, the Cytb haplotypes and GH alleles

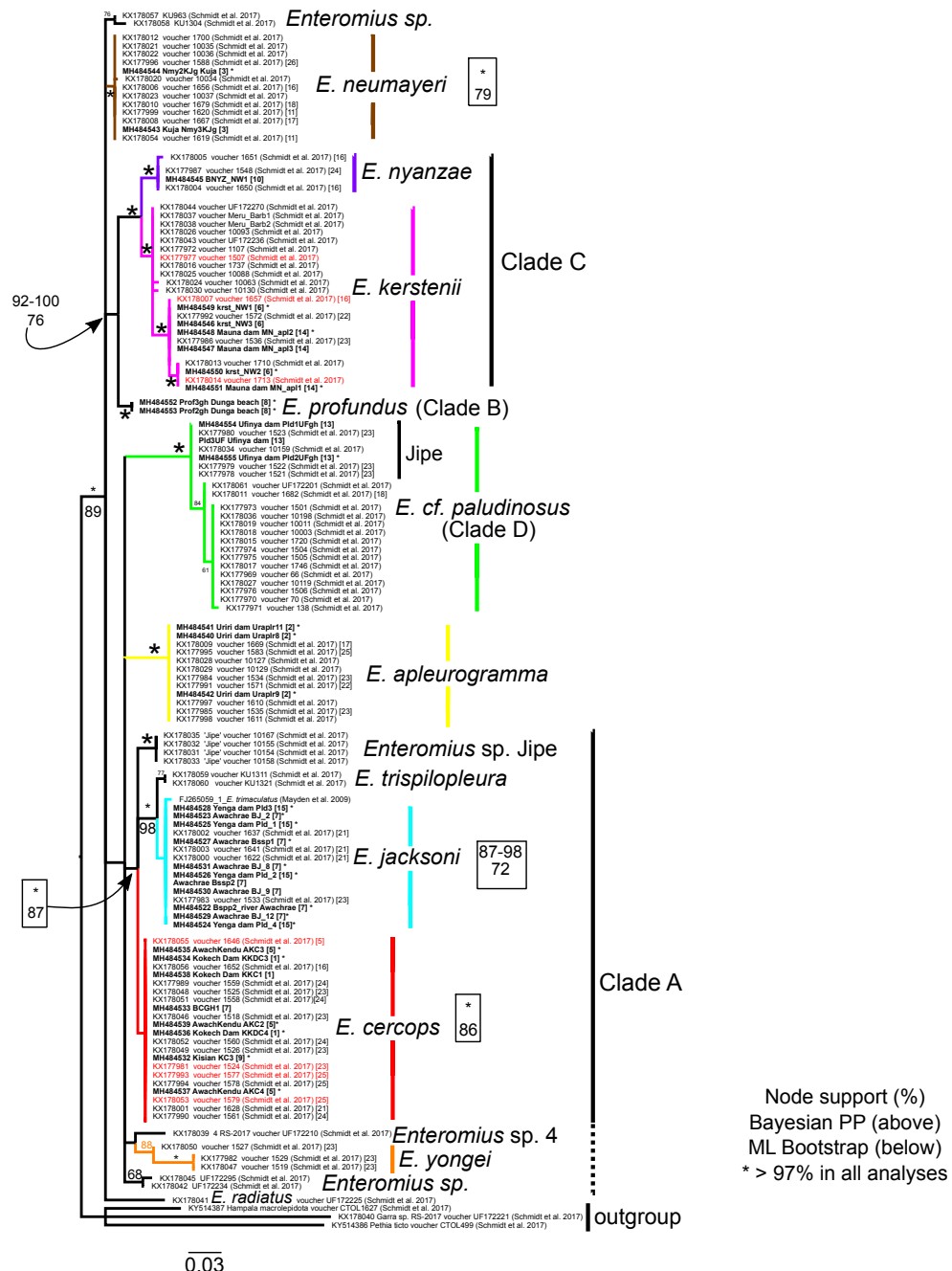

**Figure 2** **Inferred relationships based on the GH gene.** RAxML bootstrap consensus (60% majority rule) tree. Bold-faced taxon labels indicate sequences generated by the present study. Asterisks by taxon names indicate Cytb sequence was also generated in the present study. Numbers in brackets correspond to localities in Fig. 1. Node support values from Bayesian (above) and ML below are given by nodes; an asterisk indicates support >97% in all corresponding analyses. For visual clarity, several node support labels have been omitted. Red taxon labels indicate specimens identified by *Schmidt, Bart & Nyingi (2017)* as introgressed. Clade labels and colors correspond to those in other figures. The dashed line of Clade A indicates those lineages that were not found monophyletic with the rest of Clade A members (as defined by the Cytb tree; Figs. 3 and 4).

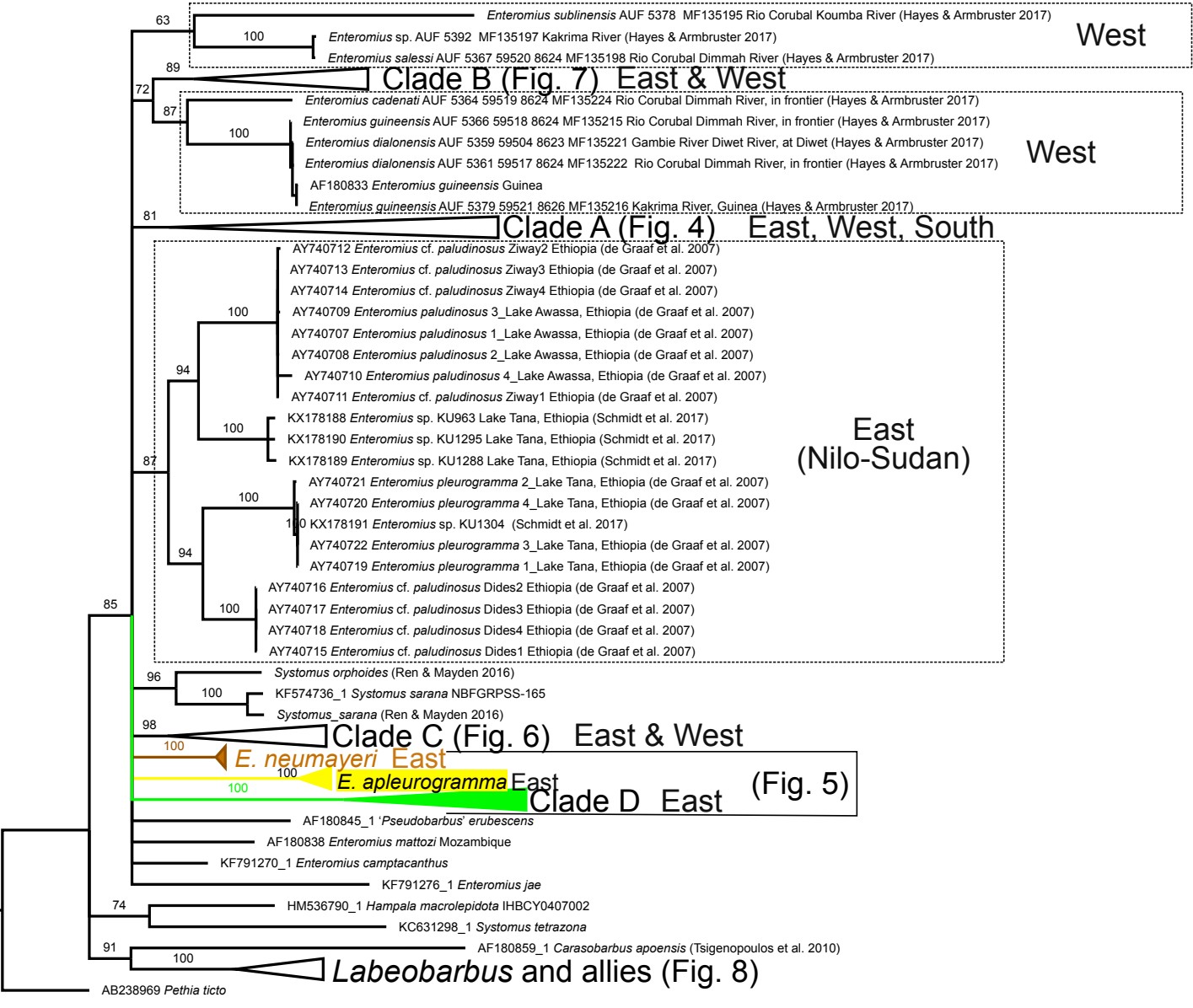

**Figure 3 Inferred relationships among the major clades in this study based on the Cytb gene, and general distribution (East, West, or South Africa; where applicable).** RAxML Bootstrap consensus (60% majority rule). Specific clades are expanded in Figs. 4–8. Numbers by nodes represent ML Bootstrap support values. For visual clarity, several node support labels have been omitted. Each taxon label contains the GenBank accession number and/or the citation and voucher ID.

that we obtained were identical or very similar to those reported by *Schmidt, Bart & Nyingi (2017)* (Figs. 2–8).

Comparison of Cytb and GH trees based on our sequences alone does not suggest instances of introgression or hybridization. All of the eight putative *E. cercops* specimens examined (localities 1, 5, 9; Fig. 1) had a GH sequence identical to the single allele reported by *Schmidt, Bart & Nyingi (2017)* for 15 *E. cercops* specimens in the same area

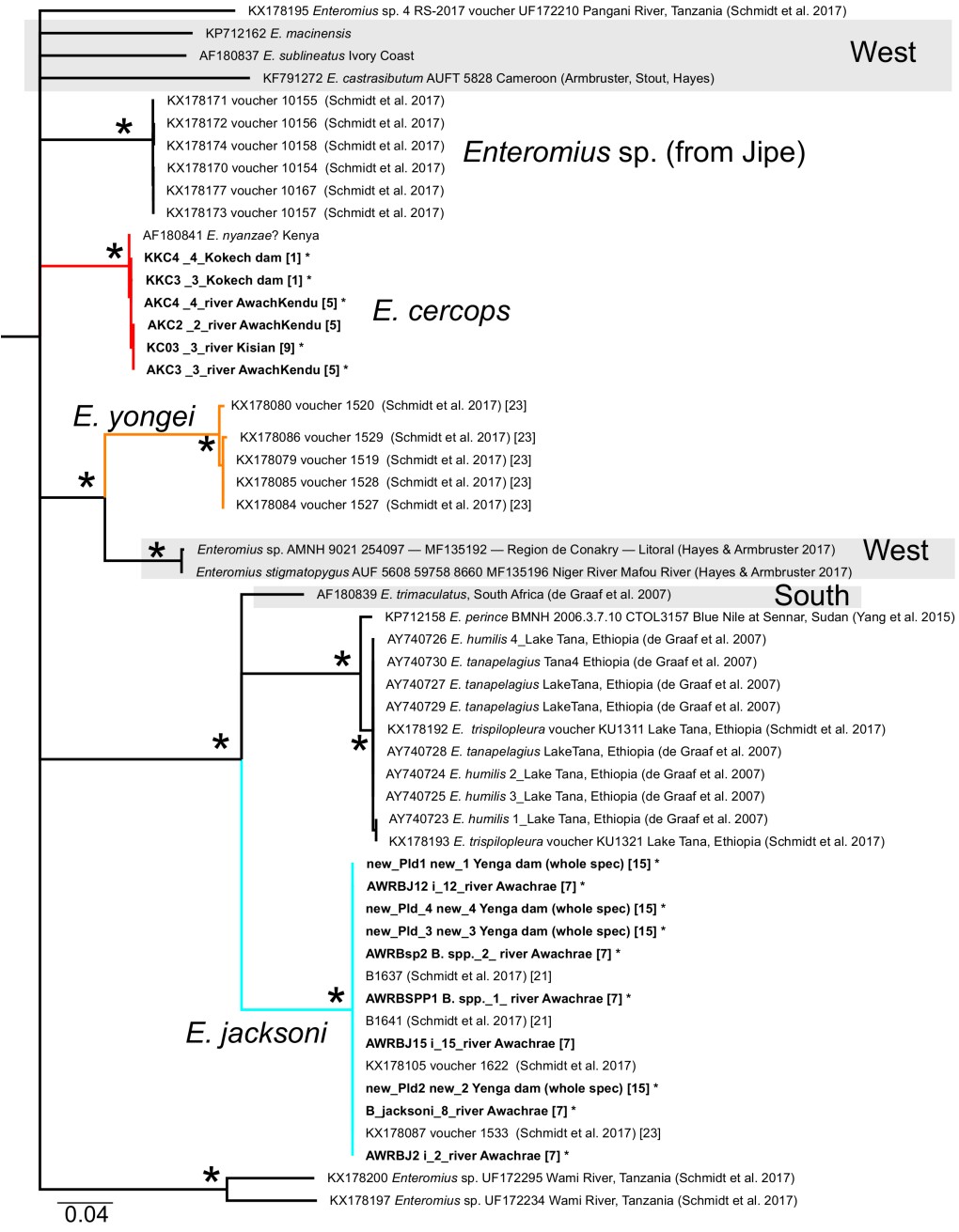

**Figure 4** **Inferred relationships within Clade A (expanded from Fig. 3) based on the Cytb gene.** RAxML bootstrap consensus (60% majority rule) tree. All taxa except those with grey shading are found in East Africa. Each taxon label contains the GenBank accession number and/or the citation and voucher ID, as well as locality label (if they were from the Lake Victoria Drainage (LVD); in bracket) corresponding to labels in Fig. 1. Bold-faced taxon labels indicate sequences generated by the present study. Asterisks by taxon names indicate GH sequence was also generated in the present study. Asterisks by nodes represent support values >97% for all analyses.

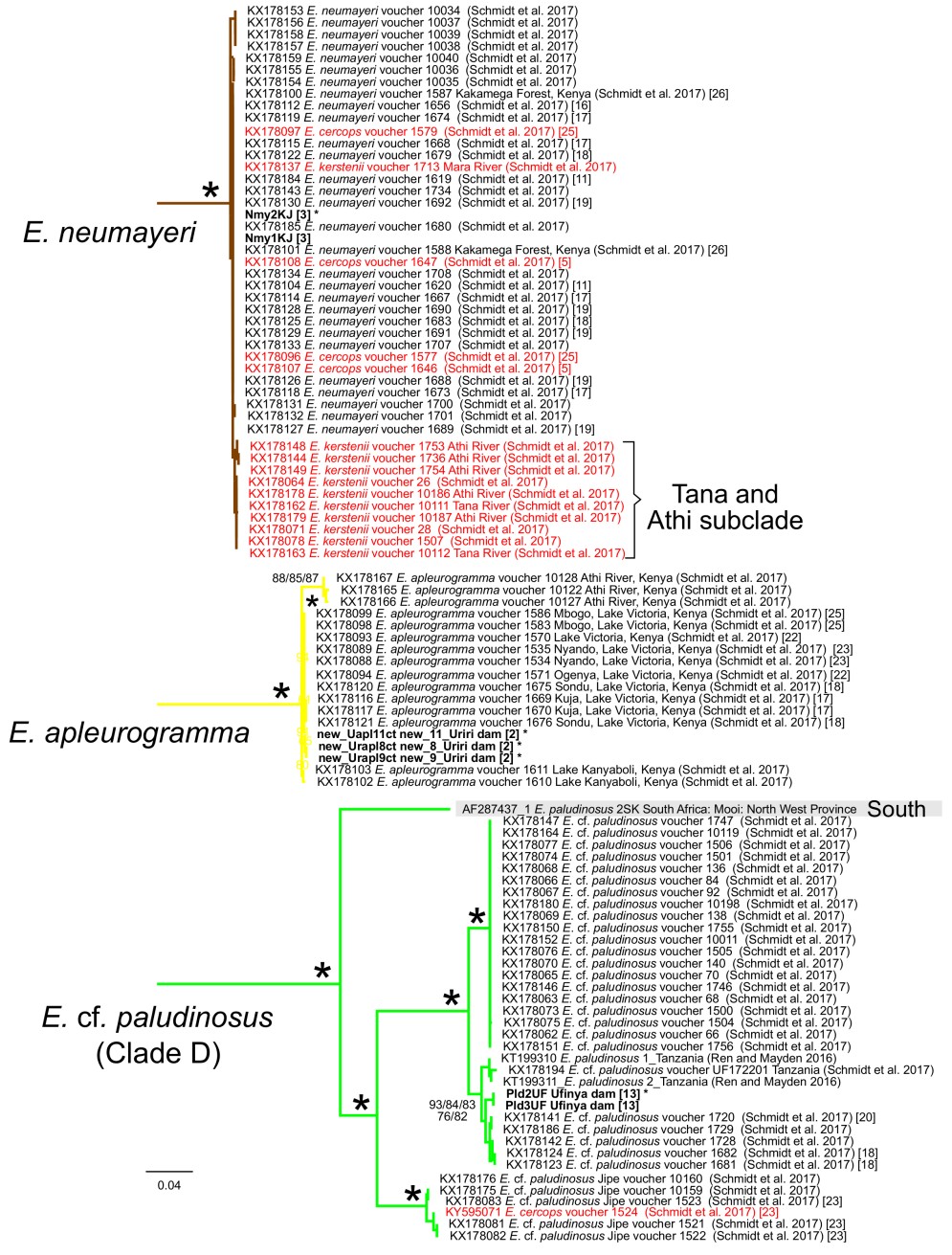

**Figure 5** **Inferred relationships within the clades of *Enteromius neumayeri*, *Enteromius apleurogramma*, and *Enteromius* cf. *paludinosus* (Clade D) (expanded from Fig. 3) based on the Cytb gene.** RAxML bootstrap consensus (60% majority rule) tree. All taxa except the one with grey shading are found in East Africa. Bold-faced taxon labels indicate sequences generated by the present study. Asterisks by taxon names indicate GH sequence was also generated in the present study. Numbers in brackets correspond to localities in Fig. 1. Node support values from Bayesian (above) and ML below are given by nodes; an asterisk indicates support >97% in all corresponding analyses. For visual clarity, several node support labels have been omitted. Red taxon labels indicate putatively introgressed individuals from *Schmidt, Bart & Nyingi (2017)*.

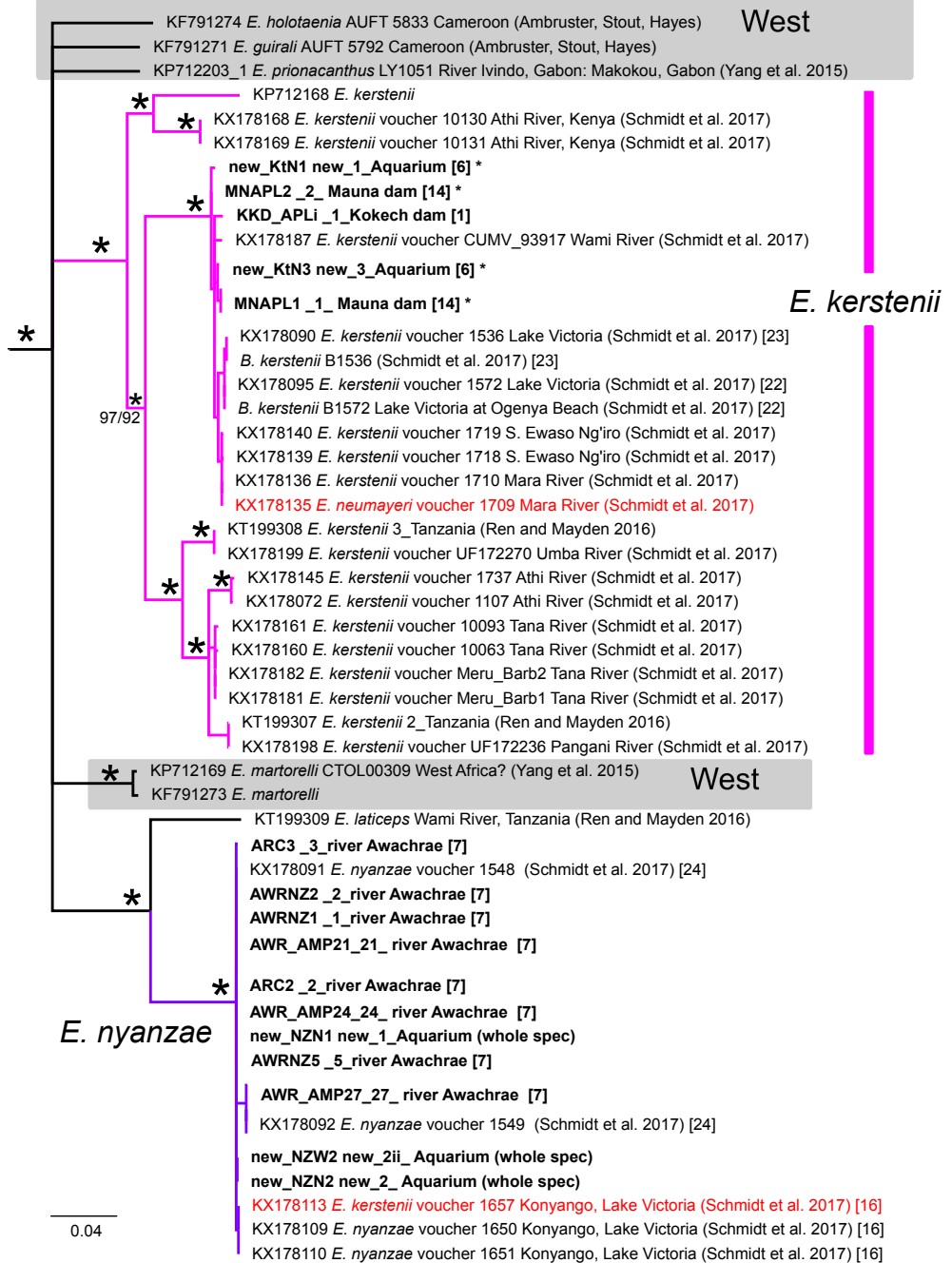

**Figure 6** **Inferred relationships within Clade C (expanded from Fig. 3) based on the Cytb gene.** All taxa except those with grey shading are found in East Africa. Bold-faced taxon labels indicate sequences generated by the present study. Asterisks by taxon names indicate GH sequence was also generated in the present study. Numbers in brackets correspond to localities in Fig. 1. Node support values from Bayesian (above) and ML below are given by nodes; an asterisk indicates support >97% in all corresponding analyses. For visual clarity, several node support labels have been omitted. Red taxon labels indicate putatively introgressed individuals from *Schmidt, Bart & Nyingi (2017)*.

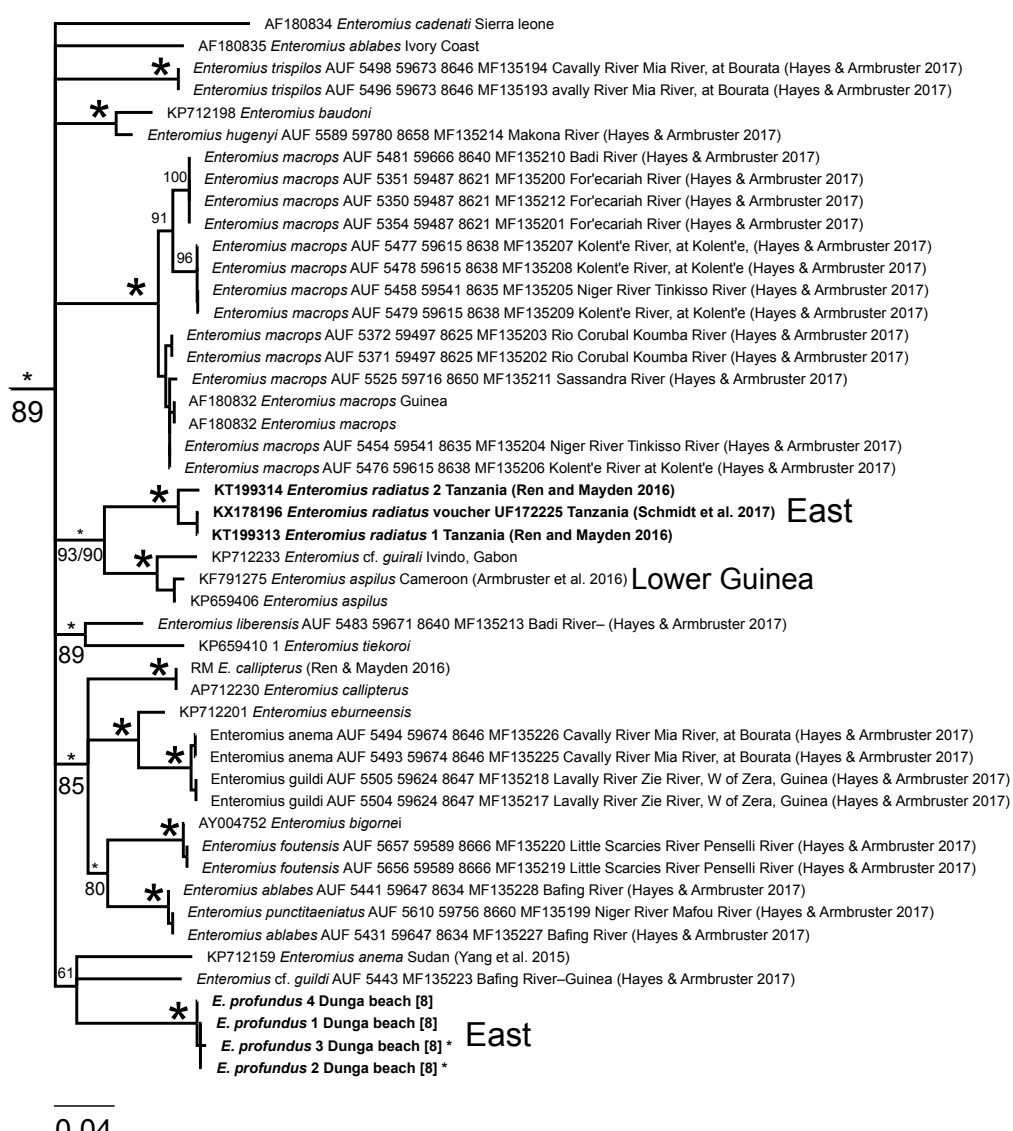

**Figure 7** **Inferred relationships within Clade B (expanded from Fig. 3) based on the Cytb gene.**
RAxML bootstrap consensus (60% majority rule) tree. All taxa except *E. radiatus* and *E. profundus*
(boldface taxon labels) are distributed in West Africa. Asterisks by taxon names indicate GH sequence was
also generated in the present study. Numbers in brackets correspond to localities in Fig. 1. Node support
values from Bayesian (above) and ML below are given by nodes; an asterisk indicates support >97% in all
corresponding analyses. For visual clarity, several node support labels have been omitted.

(i.e., localities 5, 16, 21, 23–25). The *E. cercops* GH allele is distinct from alleles found
in specimens assigned to all other species examined to date. Based on GH, the *E. cercops*
lineage (red branches in Fig. 2) forms a monophyletic group with *Enteromius* sp. Jipe
and a clade comprised of *E. jacksoni* (including one specimen assigned to *E. trimaculatus*;
FJ265059) and *E. trispilopleura* (Fig. 2).

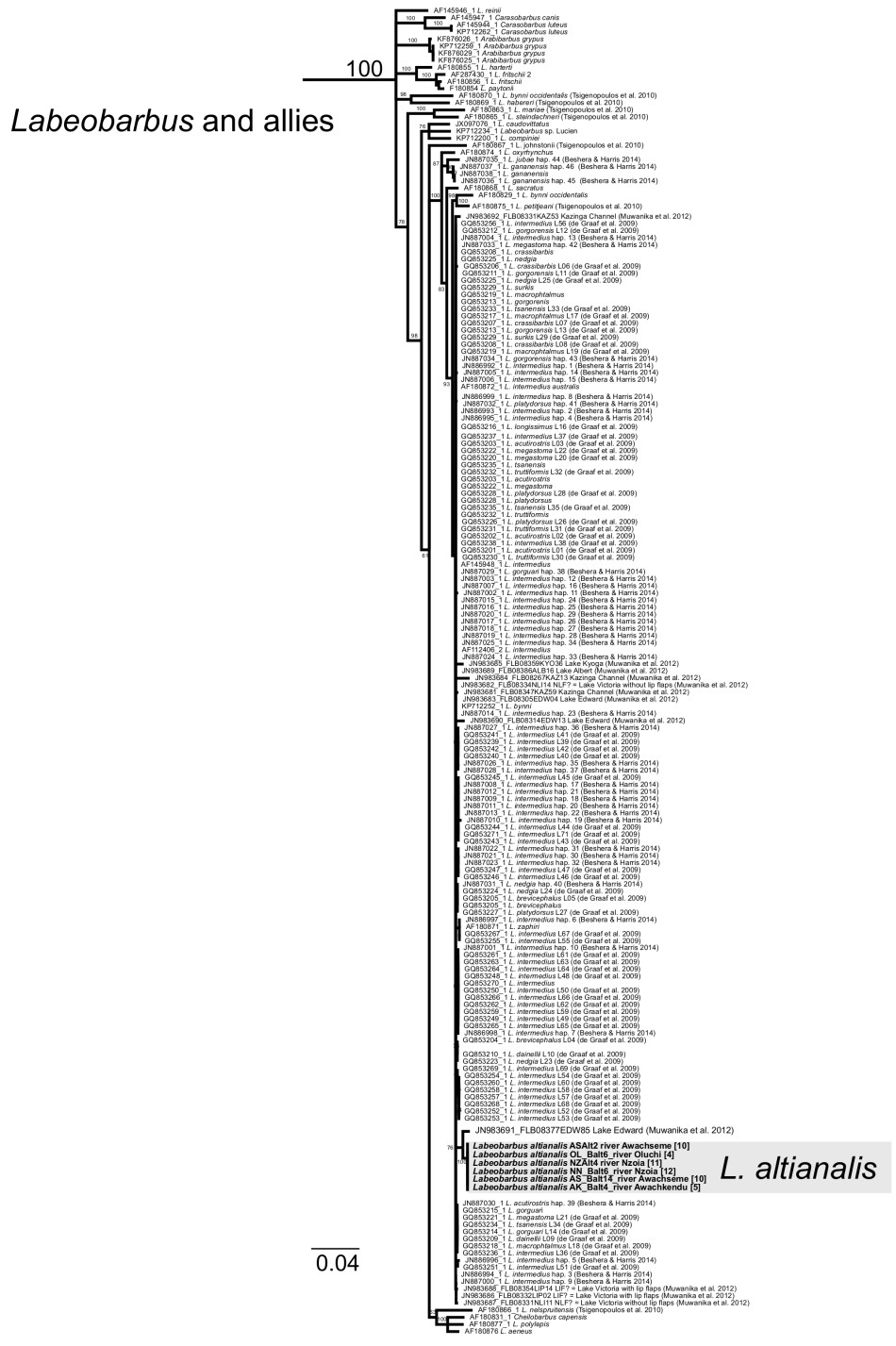

**Figure 8   Inferred relationships within the clade _Labeobarbus_ and allies (expanded from Fig. 3) based on the Cytb gene.** RAxML bootstrap consensus (60% majority rule) tree. Boldfaced taxon labels correspond to our specimens of _L. altianalis_. Numbers in brackets correspond to localities in Fig. 1. Genera names follow (_Skelton, Swartz & Vreven, 2018_; _Vreven et al., 2016_).

The Cytb sequences of our *E. cercops* specimens ($n = 6$) formed a distinct lineage (red branches in Fig. 4) that excluded the five *E. cercops* specimens reported by *Schmidt, Bart & Nyingi (2017)* and all other reported sequences to date (with the exception of GenBank record AF180841; identified as *E. nyanzae* from Kenya (*Tsigenopoulos et al., 2002*); discussed below). Maximum Cytb divergence within this clade was 0.39% (K2P). This *E. cercops* Cytb lineage was part of a larger clade (Clade A; Figs. 3 and 4) that included the closest relatives of *E. cercops* according to the GH gene (see above), as well as additional lineages assigned to several other species. The Cytb sequence of the five putative *E. cercops* specimens examined by *Schmidt, Bart & Nyingi (2017)*, including four for which they also obtained GH sequences, clustered with the Cytb sequences of other species. Four of their *E. cercops* Cytb sequences clustered with a clade made up mostly of *E. neumayeri* specimens (brown branches; Figs. 3 and 5; found in localities 5 and 25; Fig. 1), whereas one of their *E. cercops* Cytb sequences (from locality 23) clustered with individuals belonging to a subclade (i.e., Jipe) of specimens assigned to *E.* cf. *paludinosus* (Clade D; green branches; Figs. 3 and 5). Therefore, whereas *Schmidt, Bart & Nyingi (2017)* detected evidence consistent with mitochondrial introgression from other species (i.e., *E. neumayeri* and *E.* cf. *paludinosus*) into all five of the *E. cercops* specimens that they characterized for Cytb as well as for morphology or GH, we found no evidence of introgression among our specimens. Whether GenBank record AF180841 (Fig. 4) is an error or a result of introgression of *E. cercops* mitochondria into *E. nyanzae* cannot be determined (although *Schmidt, Bart & Nyingi (2017)* did not detect such introgression), because information about the nuclear genetic background is not available for this specimen.

*Schmidt, Bart & Nyingi (2017)* also detected a pattern suggestive of introgression in *E. kerstenii*. They assigned 30 specimens to *E. kerstenii* based on morphology, and characterized 16 of these for GH and 28 for Cytb (15 for both genes). The 16 GH sequences grouped into a distinct clade that excluded specimens assigned to other species (Fig. 2). In contrast, for Cytb, one specimen (KX178113) had an identical haplotype to two specimens from the same locality assigned to *E. nyanzae* (purple in Fig. 6), suggestive of recent hybridization/introgression. Similarly, the Cytb sequence of 11 specimens assigned to *E. kerstenii* grouped within the *E. neumayeri* clade (brown in Fig. 5): one of these specimens (KX178137; from the Mara River; within the LVD) had an identical Cytb haplotype to several specimens identified as *E. neumayeri* from LVD, suggesting recent or ongoing hybridization; whereas the remaining ten specimens (from the Athi and Tana rivers, which are geographically isolated from the LVD) formed a distinct, albeit shallow, subclade (Fig. 5). Because no specimens assigned to *E. neumayeri* based on morphology or a nuclear gene have been reported from the Athi or Tana drainages, it is not possible to determine whether these ten putative hybrid/introgressed *E. kerstenii* specimens are the result of recent or historical hybridization with *E. neumayeri*. In other words, it is unknown whether the *E. neumayeri* populations from the Athi or Tana drainages have diverged from those in the LVD. The Cytb sequences of the remaining 17 specimens assigned to *E. kerstenii* based on morphology (and some also based on GH) formed a highly distinct clade that excluded specimens assigned to other species (magenta in Fig. 6). This clade presumably represents pure *E. kerstenii* specimens, as its phylogenetic position (i.e., as a close relative

of *E. nyanzae*) is generally congruent between the two genes. Based on this criterion, the four *E. kerstenii* specimens for which we obtained both GH and Cytb sequences represent pure individuals.

Why the *Schmidt, Bart & Nyingi (2017)* study detected evidence of hybridization among *Enteromious* spp. in the LVD (and nearby drainages) whereas we did not, is not clear. One possible explanation is that the occurence of individuals of hybrid origin is real but rare or geographically limited, possibly at the microhabitat level, and we failed to obtain any such individuals. Alternatively, human error might have led to misidentification, mislabeling or cross-contamination of DNA template or specimens at various stages in the process, resulting in non-hybrid individuals being spuriously identified as hybrids. We consider it highly unlikely that human error would have led us to identify hybrid individuals as non-hybrids for the following reasons. First, we did not rely on morphology for evaluating evidence of introgression or hybridization. Secondly, we minimized the possibility of cross-contamination by utilizing sterile practices and by including negative controls in all of our PCR reactions. Thirdly, PCR and sequencing was repeated and confirmed for multiple of our specimens. Finally, it is unlikely that cross-contamination and/or mislabeling would have led us to infer that a hybrid individual was pure.

Our study is the first to report Cytb and GH DNA sequences from *E. profundus*, a species endemic to Lake Victoria (*Greenwood, 1970*). Each of the four specimens examined had a different Cytb haplotype (max. within clade divergence = 0.59% K2P), and formed a well-supported clade (Fig. 7). The *E. profundus* lineage falls within Clade B, which contains mostly species distributed in the Nilo-Sudan (North) and Upper Guinea (West) provinces, as well as *E. radiatus*. A sister relationship between *E. profundus* and *E. radiatus*, however, was not recovered in our Cytb or GH analyses, despite the fact that these two species had been previously synonymized (*Greenwood, 1970*) and co-occur in Lake Victoria, albeit at different depths (*Stewart, 1977*). Instead, the Cytb phylogenetic reconstruction (Fig. 7) shows a sister relationship between *E. radiatus* and a lineage comprised of specimens from West Africa (in the Lower Guinea ichthyological province) assigned to *E. aspilus* (from Cameroon) and *E.* cf. *guirali* (from Gabon). This is congruent with the results of *Ren & Mayden (2016)*, who examined the same sequences for these taxa, but lacked *E. profundus*. Therefore, *E. profundus* and *E. radiatus* do not constitute sister taxa, and phylogenetic analyses with additional taxa and markers are necessary to identify their closest relatives.

By including sequences from independent studies, our Cytb analyses revealed previously unknown relationships involving LVD species. *Enteromius yongei* was sister (∼12% K2P divergent) to a lineage comprised of two almost identical haplotypes from Guinea identified as *Enteromius sp.* and *Enteromius stigmatopygus* in *Hayes & Armbruster (2017)*, which implies an East vs. West Africa divergence (Fig. 4). Another sister relationship involving LVD taxa uncovered by our analyses was that of *E. nyanzae* (LVD) and *E. laticeps* (from Tanzania) (Fig. 6), which were ∼9% divergent (K2P).

Several interesting broad-scale phylogeographic patterns emerge with the available Cytb data on African small barbs. Continental Africa is divided into nine ichthyological provinces (reviewed in *Levêque et al., 2008*). The Lake Victoria Drainage belongs to the East Coast province. From our analyses, the following East Coast vs. West splits can be

inferred: (1) *E. radiatus* (LVD) vs. *E.* cf. *guirali* + *E. aspilus* (West: Lower Guinea province); (2) *E. profundus* (LVD) vs. one or more of the members of Clade B (all are from western Africa except *E. radiatus*); and (3) *E. yongei* (LVD) vs. *E. stigmatopygus* (West: Niger River). One or more additional East vs. West splits will be identified within clades A and C once relationships within these are resolved. Two East Coast vs. Southern ecoregion splits are inferred: (1) *E. trimaculatus* (South Africa) vs. *E. jacksoni* + *E. perince* + *E. trispilopleura* (including specimens assigned to *E. tanapelagius* and *E. humilis*; Fig. 4); and (2) the basal split within Clade D (*E.* cf. *paludinosus*). The multiple divergences between the East Coast and other provinces suggest that the dynamic and complex geological history of Africa provided opportunities, through hydrological connections, for exchange of lineages from different regions (*Salzburger, Van Bocxlaer & Cohen, 2014*; *Stewart, 2001*).

A single Cytb haplotype was recovered among the six *Labeobarbus altianalis* individuals examined (Fig. 8), representing five localities. The lack of Cytb diversity among our *L. altianalis* specimens sharply contrasts with a previous study of this species in this area that reports high haplotype diversity for the mitochondrial control region (*Chemoiwa et al. (2013)*. Although our phylogenetic analyses provide little resolution within *Labeobarbus*, they suggest that our *L. altianalis* haplotype is most similar to GenBank record JN983691 (627 bp; K2P distance = 1.2%); a specimen from Uganda (Lake Edward; Albertine drainage; ~200 km West of Lake Victoria) contributed by *Muwanika et al. (2012)*. The other *Labeobarbus* specimens examined by *Muwanika et al. (2012)* from LVD and the Albertine drainage in Uganda were 2.2–4.5% (K2P) divergent from our *L. altianalis* haplotype. *Banister (1973)* proposed, based on morphology, two groups within *Labeobarbus*: the *Labeobarbus intermedius* complex (*L. intermedius*, *L. altianalis*, *L. acuticeps*, and *L. ruasae*) and the *Labeobarbus bynni* complex (*L. bynni*, *L. gananensis*, *L. oxyrhynchus*, and *L. longifilis*). Cytb phylogenetic reconstructions in this study, however, do not support the separation of these two groups, which is congruent with the findings of a previous phylogenetic analysis that lacked *L. altianalis* sequences (*Beshera, Harris & Mayden, 2016*). Nonetheless, Cytb may be too conserved to adequately assess these relationships.

## CONCLUSION

The taxonomy and evolutionary history of the African barbs, including the role of hybridization, is far from resolved, and will require a much broader sampling of taxa, geographic locations, and genetic markers than what is presently available. Nonetheless, our analyses, which included most (if not all) available Cytb and GH sequences for this group, revealed several key insights. First, apparently pure *E. cercops* individuals do occur at the three localities where we obtained this species, including the Kendu Bay area (locality 5), where *Schmidt, Bart & Nyingi (2017)* reported a *E. cercops* specimen harboring a *E. neumayeri* mitochondrion. Secondly, *E. radiatus* does not appear to be sister to *E. profundus*, with which it was previously synonymized. Thirdly, we found evidence of several sister relationships between taxa from the East Coast and other ecoregions of Africa, suggesting that past drainage connections and vicariant events contributed to the diversification of this group. Finally, only a single haplotype was recovered among the

*L. altianalis* individuals examined, which is most similar to a specimen from Lake Edward than to specimens from other localities in Uganda.

## ACKNOWLEDGEMENTS

The Kenya Marine and Fisheries Research Institute (KMFRI) provided the samples analyzed in this study. James Woolley provided suggestions and comments regarding this work. Cecilia Smith at Texas A&M University Libraries helped with the elaboration of the study area map. This study was conducted in partial fulfillment of M.Sc. requirements (VN) at Texas A&M University.

### Funding

The M.Sc. studies of VN were supported by fellowships from Fulbright and the American Association of University Women. The Texas A&M Libraries Open Access to Knowledge (OAK) Fund covered the publication fee of this article through Basic Memberships for the authors. The funders had no role in study design, data collection and analysis, decision to publish, or preparation of the manuscript.

### Grant Disclosures

The following grant information was disclosed by the authors:
Fulbright and the American Association of University Women.
The Texas A&M University Libraries Open Access to Knowledge (OAK).

### Competing Interests

The authors declare there are no competing interests.

### Author Contributions

- Violet M. Ndeda conceived and designed the experiments, performed the experiments, analyzed the data, contributed reagents/materials/analysis tools, prepared figures and/or tables, authored or reviewed drafts of the paper, approved the final draft.
- Mariana Mateos and Luis A. Hurtado conceived and designed the experiments, analyzed the data, contributed reagents/materials/analysis tools, prepared figures and/or tables, authored or reviewed drafts of the paper, approved the final draft.

### Animal Ethics

The following information was supplied relating to ethical approvals (i.e., approving body and any reference numbers):

The Kenya Marine and Fisheries Research Institute; an official government institution of Kenya (see http://www.kmfri.co.ke/index.php/about-us/mandate-of-the-institute) provided the specimens for this research.

## Data Availability

New DNA sequences generated as part of this study are available at GenBank, accession numbers: MH484522–MH484603. The DNA sequence alignments, including the above sequences, are available as Datasets S1 and S2.

## Supplemental Information

Supplemental information for this article can be found online at http://dx.doi.org/10.7717/peerj.5762#supplemental-information.

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
