# Peer review of "Evolution of African barbs from the Lake Victoria drainage system, Kenya"

_PeerJ, doi:10.7717/peerj.5762_

## Round 0.1 · original submission · Minor Revisions

Two reviewers have recommended that this paper should be considered for publication after minor revision. I agree with them. However, please consider very seriously each of the reviewer comments, in particular reviewer 1's comments about the fact that you found no introgression while an earlier study did. this seems strange.

Also please address the order of your objectives in the introduction and discussion and make them consistent.also please note that reviewer 2 has attached ah annotated PDF with further suggestions.

Reviewer 1 ·

Basic reporting

no comment

Experimental design

no comment

Validity of the findings

no comment

Additional comments

Overall this paper follows a standard approach to addressing phylogenetic questions and it mostly does a good job. The writing is mostly clear and easy to follow. The two limitations to the study are having only two genes, but a 3-5 years ago this would have been quite good! The second is the sampling is a little bit limited, but it builds on some broader previous work, thus it is able to related their results back to the broader picture of Barbus in that region of Africa.

My one larger question that I feel the paper has not addressed is why did previous studies, especially Schmidt et al find so much introgression, including in samples from your study region, yet you did not? You don’t really discuss this point much, although you do emphasise that you didn’t find introgression, but Schmidt did. Are these species often sympatric? Any study can get some identifications incorrect, but presumably they can’t all be wrong? (I didn’t look at Schmidt et al, but presume they found discordance between mitochondrial and nuclear DNA results? If so, suggested identifications are ok if nuclear markers normally were consistent with their identifications). Presumably the introgression is older, rather than ongoing today? If so the contrast in your findings is perhaps a bit odd. I don’t know what the answers are, but you should include some discussion of this incongruence between your studies.

Most of my more minor comments are below in the order they appear in the manuscript.

Introduction

Line 48, approx. how many species of barbs / cyprinids in Africa? You do mention a number lower down, but I’m left wondering for a little bit.

Line 77 and 81, barb, not barbs (fix in the abstract too)

Line 118, rewrite sentence to improve English

Methods

Why was Catostomidae used as an outgroup initially? Is that based on some previous phylogenetic work that suggest they are sister groups?

How did you deal with heterozygous positions in growth hormone? I presume you used standard ambiguity codes? Should mention this.

You have not mentioned how you calculated your K2P distances anywhere.

Results and Discussion

You should align your main objectives at the end of the introduction (line 123) with the order that you present them in the discussion (or visa versa). For instance, your first key objective listed is the last one you address in the discussion.

You don’t provide the models chosen by partionfinder or jmodeltest for either gene, you also don’t give any indication of what partition schemes you explored under partition finder, did you try it by codon position, or just a single partition? I eventually see that you included these details in Table 1, make it clearer in line 184 what is in Table 1, i.e., best models found and partition schemes tested are provided in Table 1.

When you present some of your K2P distances, I think you are off by one decimal place, e.g., line 222, Maximum Cytb divergence within this clade was 0.39% (K2P), this should be 3.9% I think. The scale bar on the tree is 0.04, which is equivalent to 4% sequence divergence.

Line 257, max. within clade divergence = 0.59% K2P. I don’t think this statement is correct, there is almost no within clade divergence in ‘B.’ profundus. I think you mean between clade divergence to it’s nearest sister group perhaps? Unclear.

Line 278, make it clear at the start which gene you are talking about, you currently just say haplotype. How can you be sure that the Schmidt specimen is misidentified vs introgressed?

BTW, it should be RAxML, in many cases you have written RaxML which is what word autocorrects it to.

Shouldn’t start a sentence with an abbreviated genus, spell it out in full.

·

Basic reporting

Clear language – yes
Intro and background to show context, literature well referenced: Yes
Structure conforms to PeerJ – yes
Figures relevant, high quality, etc. – mostly. The map figure needs work. I don’t know why the lake is a different color. Many of the river lines do not connect? Are they endorheic or is something missing in the GIS layer?
Raw data supplied: yes

Experimental design

Original research in scope – yes
Research question well defined – yes
Rigorous investigation performed to a high technical and ethical standard – yes
Methods described in sufficient detail – yes

Validity of the findings

Impact and novelty – largely a continuation of other small barb studies, but a welcome contribution to an area of the taxonomy that has not been well-studied.
Data is robust: yes
Conclusions well stated – yes

Additional comments

This is a good paper that continues the work on the small barbs of Africa. It is well done, and a welcome addition to the literature of the group. I have made some comments and suggestions throughout the pdf. That said, the use of ‘Barbus’ is outdated. True Barbus is not even in the same tribe as the African ‘Barbus’. The only group still potentially advocating for ‘Barbus’ is Schmidt and Bart, and I even reviewed a paper by them recently using Enteromius. This is also confusing because, at one time, Labeobarbus was also referred to as Barbus in quotation marks and it is in yet another tribe as Barbus and Enteromius. The African Small Barbs will never be Barbus, and continuing to refer to them as such when there is a valid alternative is going backwards, not forwards. Not only is the use of Barbus in quotation marks backwards, the typesetting of it is difficult as well demonstrated in this paper. The Barbus part is supposed to be in italics, but the quotes are not. Other technical issues include: you have to spell out the genus name when using ‘sp.’ and the sp. is not to be italicized. The species listed as cf. guirali is probably aspilus. We did not find the two in the same clade, but they are easily confused. Make sure that you consult with Vreven et al. (2016) about Labeobarbus and Skelton et al (2018) about Pseudobarbus. Also note that these two reviews of African barbs by the top systematists working on African cypriniforms use Enteromius.

The map also needs some work. The lake should be the same color as the rivers, and it looks like at least some of the rivers that do not connect into anything are likely not endorheic.

Jon Armbruster

Skelton, P. H., E. R. Swartz and E. J. Vreven 2018 (2 Mar.) [ref. 35835] See ref. online
The identity of Barbus capensis Smith, 1841 and the generic status of southern African tetraploid cyprinids (Teleostei, Cyprinidae). European Journal of Taxonomy No. 410: 1-29.

Vreven, E. J. W. M. N., T. Musschoot, J. Snoeks, and U. K. Schliewen. 2016. The African hexaploid Torini (Cypriniformes: Cyprinidae): review of a tumultuous history. Zoological Journal of the Linnean Society 177:231–305.

---

## Round 0.2 · accepted · Accept

I think the authors have satisfactorily addressed the issues raised by reviewers and the paper is now suitable for publication.

#